# Geochemical Characteristics of Alluvial Aquifer in the Varaždin Region

Igor Karlović [1], Tamara Marković [1,*], Martina Šparica Miko [1] and Krešimir Maldini [2]

[1] Croatian Geological Survey, 10 000 Zagreb, Croatia; ikarlovic@hgi-cgs.hr (I.K.); mtsparica@hgi-cgs.hr (M.Š.M.)
[2] Croatian Waters, Central Water Management Laboratory, 10 000 Zagreb, Croatia; Kresimir.Maldini@voda.hr
[*] Correspondence: tmarkovic@hgi-cgs.hr

**Abstract:** The variation in the major groundwater chemistry can be controlled by dissolution and precipitation of minerals, oxidation-reduction reactions, sorption and exchange reactions, and transformation of organic matter, but it can also occur as a result of anthropogenic influence. The alluvial aquifer represents the main source of potable water for public water supply of the town Varaždin and the surrounding settlements. Sampling campaigns were carried out from June 2017 until June 2019 to collect groundwater samples from nine observation wells. Major cations and anions, dissolved organic carbon and nutrients were analyzed in the Hydrochemical Laboratory of Croatian Geological survey. The sampled waters belong to the CaMg-HCO$_3$ hydrochemical type, except the water from observation well P-4039 that belongs to NaCa-HCO$_3$ hydrochemical type. It was identified that groundwater chemistry is mainly controlled by hydrogeological environment (natural mechanism), but anthropogenic influence is not negligible. The results of this research have significant implications on sustainable coexistence between agricultural production and water supply.

**Keywords:** shallow alluvial aquifer; major cations and anions; nutrients; Croatia





## 1. Introduction

Natural waters acquire their chemical characteristics both by dissolution and by chemical reactions with solids, liquids and gases, with which they come into contact during the various phases of the hydrological cycle [1]. The chemical composition of groundwater is often used to investigate groundwater residence time, origin, flow direction and anthropogenic or natural contamination [2–10]. The variation in the major cations and anions of groundwater can be controlled by dissolution and precipitation of minerals, oxidation-reduction reactions, sorption and exchange reactions, and transformation of organic matter. In addition, major cations and anions can be added to the aquifer systems as a result of anthropogenic influence; for example calcium, magnesium, sodium, chloride and potassium are present in sludge, waste water, manure [11–13].

In recent decades, high nitrate concentrations emerged as a globally growing problem for drinking and agricultural purposes [14–16]. The adverse health effects of high nitrate levels in drinking water have been well documented, including gastric cancer, non-Hodgkin's lymphoma, and methemoglobinemia [17–19]. In some parts of Croatia, uncontrolled and extensive agricultural production is causing the pollution of groundwater with nitrate. An example of an area where high nitrate concentrations are present is the alluvial aquifer in the northwestern part of Croatia, in the Varaždin region. To determine the impact of the hydrogeological environment and humans on groundwater chemical features of the alluvial aquifer, geochemical investigations were performed. The aquifer represents the main source of potable water for public water supply of the town of Varaždin and the surrounding settlements. The most important activity in the region is agricultural production (plantation of wheat, maize, cabbage, poultry and dairy farming). The present research describes the geochemical reactions that influence the chemical composition of groundwater, and evaluates the key controlling processes within the alluvial aquifer. The

main aim of the paper is to assess the natural and human influence on the groundwater chemistry in the study area, in order to assure sustainable coexistence between agricultural production and water supply, and prevent further groundwater quality deterioration.

## 2. Description of the Study Area

The study area is located in the northwestern part of Croatia, in the Varaždin region. It belongs to the Black Sea catchment area. The aquifer, which is situated in the Drava river lowland, is characterized by intergranular porosity. The topography is characterized by wide flatlands surrounded by hills. The Drava River presents the aquifer boundary in the northwest and north, and in the southeastern part of the study area, the Plitvica stream flows (Figure 1).

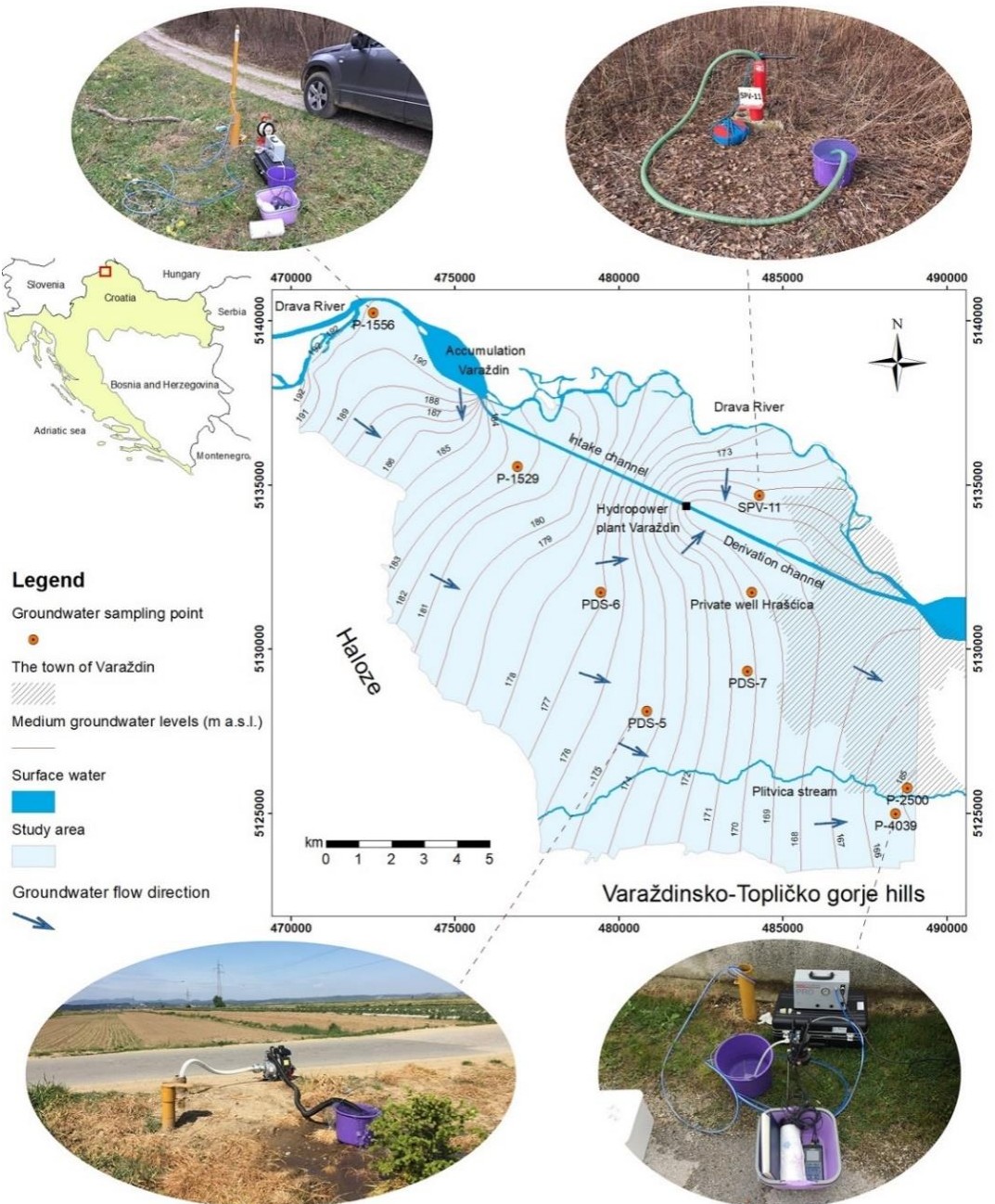

**Figure 1.** Geographical position of the study area with locations of the groundwater sampling points and photographs of pumping the observation wells at characteristic locations (P-1556, PDS-5, SPV-11, P-4039). The general groundwater flow direction is defined by the head contours for medium water levels measured on 14 October 2013.

The study area has a characteristic precipitation regime with more precipitation during the summer [20]. Consequently, the local climate is categorized in the Cfb group according to the Köppen–Geiger classification system, which is known as "warm-temperate climate" or "marine west coast climate." The study area is the former [21]. The mean annual temperature is 10.6 °C, with January as the coldest month (average temperatures of 0.0 °C), and July and August as the warmest months (average temperatures of 20.9 and 20.1 °C, respectively) [9]. According to the data from the last climate normal period (1981–2010), the average annual precipitation was 832 mm, with a lower average precipitation amount during the cold part of the year (minimum in January with 38.7 mm), and a higher average precipitation amount during the warm part of the year (maximum in September with 98.3 mm) [9]. The favorable climate, topography and available groundwater have enabled intensive agricultural practices, including the application of synthetic fertilizers and manure that has subsequently led to high nitrate concentrations in the Varaždin aquifer.

The aquifer is composed of gravel and sand with variable portions of silt [22–24]. It was formed during the Pleistocene and Holocene as a result of accumulation processes of the Drava River [25]. At the utmost northwestern part of the study area, the aquifer thickness is less than 5 m, and it gradually increases in downstream direction, reaching its maximum of roughly 50 m in the eastern part of the study area (Figure 1). It is noticed that particle size changes going from the northwestern part downstream, i.e., the size of gravel and sand particles gets gradually smaller as result of the decrease in energy of the Drava River. Deposits of gravel and sand show stratification in some places, which is characterized by a sudden change in the size of pebbles, or an increased amount of sandy component [26]. Gneiss and quartz pebbles prevail, but there are also pebbles of basic and neutral eruptive rocks; limestone, dolomite, etc. [27]. The main constituents of the sand are quartz, feldspars and carbonate minerals, and it contains significant amounts of heavy minerals such as garnet, epidote, amphibole, rutile, kyanite, etc. [28,29]. Along with gravel and sand in the study area, there are also oxbow deposits that were deposited in the old Drava riverbed, where the still water environment of sedimentation remained for a long time. Various fine sediments were deposited, such as silt, clay and organic matter, forming distinctive facies of oxbow [28,29]. Remains of oxbows have been observed in the area of Strmec, Petrijanec, Otok Virje, Svibovec and Sračinec.

In the southeastern part of the study area, near Varaždin town, a tiny aquitard composed of clay and silt appears, dividing the aquifer into two hydrogeological units. The aquitard has regional significance, especially downstream outside the study area, but not so much in the study area, due to its small thickness. The covering layer of the aquifer is not continuously developed throughout the entire study area. In the central part and near the Drava River it rarely exceeds 50 cm, while often it completely disappears. Such conditions are favorable if they are considered from the aspect of aquifer recharge, but at the same time, tiny covering layers makes the aquifer quite vulnerable. The aquifer of the study area is unconfined, and is recharged by precipitation infiltration through unsaturated zones and by surface water percolation [9,30]. The general groundwater flow direction is NW–SE and is parallel to the Drava River (Figure 1). It is noteworthy that the groundwater flow net has been significantly changed since the building of a hydroelectric power plant in 1970s. Namely, prior to this intervention, the groundwater had flown towards the Drava River, which had represented the discharge zone, and now it is the recharge zone. After the construction of the Varaždin accumulation lake, the pressure head layout changed, leading to percolation of the lake water to the aquifer. At the same time, in the vicinity of the derivation channel, the groundwater level is lowered because the channel is deeply cut into the aquifer (Figure 1). Another discharge zone is the Plitvica stream, which drains the aquifer most of the time and recharges it only in high water level conditions.

## 3. Materials and Methods

Groundwater sampling campaigns were carried out from June 2017 until June 2019. Samples were collected from alluvial aquifer by pumping 9 observation wells (8 piezometric

wells and 1 private well). The depths of observation wells and their screens (if known) are given in Table 1.

**Table 1.** Observation wells information.

| Observation Well | Elevation (m a.s.l.) | Depth of the Well (m) | Depth Interval of the Screen (m) |
|---|---|---|---|
| Private well Hrašćica | 176.00 | 15.0 | 5–15 |
| PDS-5 | 178.36 | 31.0 | 13.7–19.7 |
| PDS-6 | 184.07 | 25.0 | 11.7–17.7 |
| PDS-7 | 175.71 | 42.5 | 29.3–32.3 |
| P-1529 | 187.32 | 8.0 | n.a. [1] |
| P-1556 | 193.03 | 15.6 | n.a. [1] |
| P-2500 | 167.81 | 5.20 | n.a. [1] |
| P-4039 | 167.76 | 8.0 | n.a. [1] |
| SPV-11 | 177.69 | 40.0 | 24.5–35.8 |

[1] information about screen interval not available.

In situ parameters such as temperature, pH, dissolved oxygen (DO) and electrical conductivity (EC) were measured in the field using a WTW multi-probe. Alkalinity was also measured in the field by titration with 1.6 N $H_2SO_4$, using phenolphthalein and bromocresol green-methyl red as indicators, and then converted to the equivalent $HCO_3$ concentrations. Samples for analysis of cations and anions were filtered through 0.45 μm cellulose membrane filters into the HPDE 500 mL bottles prior measuring on Ion Chromatographer Dionex ICS 6000, while low concentrations of $NH_4^+$, $NO_2^-$ and $PO_4^{3-}$-P were analyzed using spectrophotometer HACH DR 9000. Samples for measurement of dissolved organic carbon (DOC) were, in the field, collected into 100 mL dark glass bottles and analyzed using HACH QBD1200 analyzer. Samples were kept in the portable refrigerator during transport to the laboratory and analyzed in the evening of the same day. The ion balance errors for the analyses were checked by the relative deviation from charge balance ($\Delta_{meq} = 100 \times (\Sigma_{meq+} - \Sigma_{meq-})/(\Sigma_{meq+} + \Sigma_{meq-}) < \pm5\%$) [31,32]. Concentrations of dissolved metals in water were measured using inductively coupled plasma-mass spectrometry on Agilent 8900 ICP-MS Triple Quad with solution of 30 μg $L^{-1}$ Ge, Y, In and Tb as internal standards according to HRN EN ISO 17294-2:2016 norm [33]. All measurements were performed in quintuplets. Quality control of the ICP-MS method was performed by the analysis of the elements of interest in certified reference material Anas-38 (Inorganic Venture) at the beginning and after analyzing each series of samples. Calibration lines for each element and internal standards were made using Agilent multi-element calibration standard solutions and internal standard mix solution. Before the analysis, the samples were filtered through a 0.45 μm filter on the field, and acidified with ultra-pure 6 N $HNO_3$ acid. The PHREEQC software was used to determine saturation indices and $CO_2$ pressure [34]. The determination of the redox state within the aquifer was performed using McMahon and Chapelle's methodology [35]. The correlation diagrams and calculation of correlation coefficients were determined using MS Excel tool.

## 4. Results

The average, minimum and maximum values of the analyzed physicochemical parameters and metal concentrations in the groundwater samples are presented in Table 2a,b The calculated redox conditions of groundwater are given in Table 3.

**Table 2.** (a) The average, minimum and maximum values of the analyzed physicochemical parameters in the groundwater samples. (b) The average, minimum and maximum values of metal concentrations in the groundwater samples.

**(a)**

| | | EC (µS/cm) | T (°C) | pH | $O_2$ (mg/L) | $HCO_3^-$ (mg/L) | $PO_4^{3-}$-P (mg/L) | $NH_4^+$ (mg/L) | $NO_2^-$ (mg/L) | $Cl^-$ (mg/L) | $SO_4^{2-}$ (mg/L) | $NO_3^-$ (mg/L) | TN (mg/L) | $Br^-$ (mg/L) | $Ca^{2+}$ (mg/L) | $Mg^{2+}$ (mg/L) | $Na^+$ (mg/L) | $K^+$ (mg/L) | DOC (mg/L) | $SiO_2$ (mg/L) |
|---|---|---|---|---|---|---|---|---|---|---|---|---|---|---|---|---|---|---|---|---|
| Private well | min | 673 | 10.8 | 6.91 | 3.1 | 342 | <0.01 | <0.01 | <0.01 | 9.3 | 21.0 | 41.8 | 10.6 | <0.10 | 101 | 18.9 | 7.2 | 3.9 | 0.30 | 11.4 |
| | max | 747 | 13.8 | 7.40 | 8.5 | 414 | 0.22 | 0.05 | 0.02 | 43.1 | 34.0 | 91.9 | 20.9 | 4.0 | 133 | 23.8 | 19.7 | 5.3 | 0.83 | 14.4 |
| | average | 713 | 13.1 | 7.22 | 6.7 | 382 | 0.04 | 0.02 | 0.01 | 19.0 | 26.6 | 58.4 | 14.4 | 2.5 | 111 | 20.7 | 9.2 | 4.5 | 0.45 | 12.5 |
| P-1529 | min | 755 | 10.3 | 6.86 | 1.5 | 388 | <0.01 | <0.01 | <0.01 | 14.1 | 21.0 | 40.1 | 11.2 | <0.10 | 108 | 19.9 | 13.5 | 5.1 | 0.19 | 10.1 |
| | max | 814 | 14.5 | 7.36 | 9.2 | 512 | 0.54 | 0.13 | 0.24 | 37.3 | 31.8 | 77.1 | 17.5 | 4.0 | 153 | 28.3 | 17.5 | 12.5 | 4.6 | 18.4 |
| | average | 787 | 12.6 | 7.15 | 6.1 | 437 | 0.10 | 0.05 | 0.04 | 23.1 | 25.8 | 55.7 | 14.0 | 2.9 | 119 | 24.6 | 15.1 | 6.1 | 1.3 | 12.7 |
| P-1556 | min | 658 | 9.40 | 6.93 | 0.6 | 381 | <0.01 | <0.01 | <0.01 | 5.7 | 9.4 | 5.3 | 2.2 | <0.10 | 105 | 17.2 | 5.2 | 4.2 | 0.43 | 9.2 |
| | max | 877 | 16.0 | 7.45 | 7.8 | 512 | 0.76 | 0.07 | 0.04 | 32.9 | 34.8 | 35.6 | 6.7 | <0.10 | 154 | 22.5 | 21.3 | 10.1 | 5.4 | 15.1 |
| | average | 743 | 13.1 | 7.15 | 4.9 | 442 | 0.10 | 0.03 | 0.02 | 14.7 | 23.8 | 17.4 | 4.4 | <0.10 | 123 | 19.6 | 9.9 | 5.8 | 1.2 | 11.8 |
| SPV-11 | min | 490 | 12.0 | 7.03 | 1.4 | 249 | <0.01 | <0.01 | <0.01 | 5.3 | 24.4 | 8.0 | 2.2 | <0.10 | 73.9 | 15.5 | 2.4 | 0.94 | 0.13 | 10.3 |
| | max | 496 | 14.4 | 7.56 | 7.7 | 278 | 0.14 | 0.07 | 0.02 | 24.2 | 33.2 | 18.9 | 5.00 | <0.10 | 82.6 | 20.6 | 13.1 | 3.8 | 0.60 | 16.1 |
| | average | 494 | 12.3 | 7.43 | 2.2 | 267 | 0.05 | 0.03 | 0.01 | 9.1 | 27.3 | 11.2 | 2.7 | <0.10 | 75.8 | 16.8 | 3.9 | 1.6 | 0.59 | 12.7 |
| P-2500 | min | 696 | 10.8 | 7.01 | 0.9 | 325 | <0.01 | <0.01 | <0.01 | 18.7 | 15.0 | 27.5 | 11.3 | <0.10 | 99.4 | 19.0 | 11.5 | 1.0 | 0.52 | 10.5 |
| | max | 802 | 17.9 | 7.55 | 9.2 | 405 | 0.64 | 0.16 | 0.04 | 65.6 | 46.8 | 137 | 31.3 | 5.8 | 138 | 22.7 | 48.1 | 2.9 | 1.2 | 15.6 |
| | average | 737 | 13.7 | 7.29 | 6.5 | 361 | 0.07 | 0.04 | 0.01 | 34.6 | 31.3 | 66.4 | 15.9 | 3.6 | 112 | 20.6 | 22.3 | 1.7 | 0.71 | 12.4 |
| P-4039 | min | 766 | 11.0 | 7.01 | 0.2 | 238 | <0.01 | <0.01 | <0.01 | 75.8 | 6.6 | <0.10 | <1.0 | <0.10 | 82.6 | 10.7 | 47.1 | 1.5 | 0.69 | 9.6 |
| | max | 1091 | 14.7 | 7.71 | 4.2 | 410 | 0.35 | 0.14 | 0.06 | 279 | 40.2 | 21.9 | 2.4 | 4.2 | 147 | 26.5 | 157 | 3.7 | 2.4 | 22.0 |
| | average | 975 | 13.2 | 7.40 | 1.5 | 324 | 0.08 | 0.04 | 0.02 | 170 | 28.2 | 5.1 | 1.4 | 2.8 | 107 | 21.9 | 81.5 | 2.8 | 1.3 | 13.1 |
| PDS-5 | min | 661 | 11.8 | 6.91 | 3.6 | 322 | <0.01 | <0.01 | <0.01 | 8.8 | 13.0 | 42.5 | 12.1 | <0.10 | 100 | 18.7 | 5.3 | 11.4 | 0.25 | 10.7 |
| | max | 694 | 13.9 | 7.45 | 9.9 | 456 | 0.82 | 0.07 | 0.02 | 22.4 | 67.5 | 210 | 47.7 | 3.0 | 147 | 29.9 | 15.9 | 88.8 | 0.52 | 19.7 |
| | average | 683 | 12.6 | 7.28 | 8.3 | 387 | 0.12 | 0.03 | 0.01 | 14.7 | 27.3 | 83.0 | 20.0 | 1.5 | 113 | 22.7 | 6.2 | 31.3 | 0.35 | 13.4 |
| PDS-6 | min | 708 | 11.4 | 6.92 | 7.3 | 260 | <0.01 | <0.01 | <0.01 | 9.0 | 11.0 | 38.5 | 11.8 | <0.10 | 86.3 | 17.9 | 7.0 | 1.5 | 0.30 | 10.8 |
| | max | 744 | 13.0 | 7.45 | 9.1 | 456 | 0.22 | 0.04 | 0.02 | 24.0 | 30.0 | 123 | 27.9 | 4.0 | 150 | 21.7 | 9.7 | 3.0 | 2.5 | 29.9 |
| | average | 730 | 12.4 | 7.17 | 8.7 | 381 | 0.05 | 0.03 | 0.01 | 14.7 | 22.2 | 65.4 | 15.4 | 2.5 | 120 | 19.6 | 8.0 | 2.2 | 0.61 | 14.4 |
| PDS-7 | min | 730 | 11.2 | 6.99 | 6.6 | 280 | <0.01 | <0.01 | <0.01 | 11.4 | 19.0 | 52.5 | 12.6 | <0.10 | 112 | 19.5 | 4.3 | 0.40 | 0.25 | 10.9 |
| | max | 777 | 13.0 | 7.76 | 10.9 | 435 | 1.1 | 0.04 | 0.04 | 68.0 | 35.9 | 180 | 41.0 | 4.0 | 151 | 25.4 | 15.0 | 8.9 | 2.6 | 23.5 |
| | average | 758 | 12.3 | 7.32 | 8.9 | 374 | 0.14 | 0.02 | 0.02 | 20.4 | 27.7 | 96.7 | 23.2 | 3.0 | 123 | 21.4 | 6.7 | 1.9 | 0.64 | 15.1 |

(**b**)

| | | As (µg/L) | Cd (µg/L) | Cr (µg/L) | Cu (µg/L) | Fe (µg/L) | Li (µg/L) | Mn (µg/L) | Mo (µg/L) | Ni (µg/L) | Pb (µg/L) | Sr (µg/L) | Zn (µg/L) |
|---|---|---|---|---|---|---|---|---|---|---|---|---|---|
| Private well | min | 0.11 | <0.01 | 0.28 | 0.49 | 1.1 | 0.68 | 0.08 | 0.51 | <0.02 | 0.06 | 135 | 13.2 |
| | max | 0.44 | 0.06 | 0.57 | 7.3 | 19.8 | 3.9 | 1.7 | 0.99 | 1.1 | 0.46 | 211 | 88.4 |
| | average | 0.18 | 0.02 | 0.43 | 1.40 | 6.6 | 2.1 | 0.54 | 0.77 | 0.51 | 0.20 | 181 | 40.2 |
| P-1529 | min | 0.07 | <0.01 | 0.25 | 0.55 | 2.1 | 1.2 | 0.06 | 0.36 | 0.22 | 0.11 | 171 | 28.0 |
| | max | 0.36 | 0.10 | 0.89 | 6.1 | 21.4 | 4.5 | 9.8 | 0.64 | 1.1 | 1.7 | 267 | 78.0 |
| | average | 0.16 | 0.05 | 0.45 | 2.3 | 7.3 | 2.7 | 2.7 | 0.51 | 0.52 | 0.53 | 235 | 45.9 |
| P-1556 | min | 0.34 | <0.01 | <0.02 | 0.30 | 208 | 1.2 | 0.29 | 0.33 | 0.05 | 0.10 | 242 | 104 |
| | max | 1.1 | 0.05 | 0.26 | 3.5 | 483 | 3.6 | 237 | 0.71 | 3.2 | 0.60 | 371 | 2134 |
| | average | 0.81 | 0.03 | 0.08 | 0.96 | 349 | 2.7 | 158 | 0.52 | 1.1 | 0.27 | 303 | 557 |
| SPV-11 | min | 0.10 | <0.01 | 0.06 | 0.07 | 1.7 | <0.01 | <0.05 | 0.53 | 0.12 | 0.05 | 243 | 3.5 |
| | max | 0.39 | 0.06 | 0.46 | 1.3 | 41.8 | 1.1 | 0.98 | 0.95 | 5.3 | 0.41 | 327 | 13.5 |
| | average | 0.19 | 0.02 | 0.18 | 0.57 | 9.3 | 0.69 | 0.42 | 0.78 | 0.69 | 0.17 | 296 | 6.2 |
| P-2500 | min | 0.09 | <0.01 | 0.03 | 0.15 | 1.8 | 0.14 | 0.27 | 0.12 | 0.11 | 0.05 | 126 | 7.6 |
| | max | 0.30 | 0.08 | 0.42 | 2.5 | 184 | 2.1 | 8.8 | 0.34 | 4.6 | 0.76 | 212 | 76.3 |
| | average | 0.15 | 0.02 | 0.13 | 1.0 | 19.4 | 1.3 | 1.6 | 0.21 | 0.61 | 0.19 | 172 | 18.6 |
| P-4039 | min | 0.12 | 0.01 | <0.02 | 0.13 | 135 | 0.95 | 11.1 | 0.28 | 0.06 | 0.11 | 146 | 427 |
| | max | 0.85 | 0.09 | 0.44 | 4.1 | 2265 | 2.9 | 30.5 | 0.94 | 46.7 | 4.6 | 251 | 5600 |
| | average | 0.27 | 0.03 | 0.11 | 1.1 | 744 | 1.8 | 19.8 | 0.52 | 2.6 | 1.2 | 205 | 2873 |
| PDS-5 | min | 0.11 | <0.01 | 0.37 | 0.11 | 1.5 | 0.10 | 0.29 | 0.19 | <0.02 | 0.06 | 143 | 205 |
| | max | 0.39 | 0.05 | 0.63 | 6.7 | 34.0 | 2.9 | 1.6 | 0.34 | 1.7 | 0.60 | 223 | 543 |
| | average | 0.17 | 0.02 | 0.51 | 0.69 | 8.2 | 1.6 | 0.68 | 0.26 | 0.32 | 0.19 | 188 | 330 |
| PDS-6 | min | 0.08 | <0.01 | 0.27 | 0.15 | 2.0 | 0.89 | 0.25 | 0.16 | <0.02 | 0.41 | 192 | 265 |
| | max | 0.35 | 0.11 | 0.66 | 3.2 | 9.6 | 4.2 | 4.5 | 0.71 | 0.84 | 1.7 | 309 | 627 |
| | average | 0.14 | 0.03 | 0.47 | 0.88 | 4.9 | 2.6 | 1.1 | 0.25 | 0.34 | 0.76 | 260 | 466 |
| PDS-7 | min | 0.05 | <0.01 | 0.15 | 0.02 | 10.2 | 0.31 | 2.2 | 0.29 | 0.04 | 0.01 | 116 | 169 |
| | max | 0.33 | 0.09 | 0.82 | 1.6 | 59.3 | 2.7 | 16.5 | 0.58 | 1.3 | 1.5 | 268 | 5665 |
| | average | 0.11 | 0.04 | 0.49 | 0.72 | 34.1 | 1.6 | 7.8 | 0.44 | 0.44 | 0.48 | 227 | 2169 |

**Table 3.** The calculated redox conditions of groundwater.

| Observation Well | General Redox Category | Redox Process |
|---|---|---|
| Private well Hrašćica | Oxic | $O_2$ |
| P-1529 | Oxic | $O_2$ |
| P-1556 | Mixed (oxic-anoxic) | $O_2$-Fe(III)/$SO_4$ or $O_2$-Mn(IV) |
| SPV-11 | Oxic | $O_2$ |
| P-2500 | Oxic | $O_2$ |
| P-4039 | Anoxic or Mixed (oxic-anoxic) | $NO_3$-Fe(III)/$SO_4$ or $O_2$-Fe(III)/$SO_4$ |
| PDS-5 | Oxic | $O_2$ |
| PDS-6 | Oxic | $O_2$ |
| PDS-7 | Oxic | $O_2$ |

In situ parameters are presented in Table 2a. The EC values ranged from 490 to 1091 μS/cm, with the highest values measured in water samples from P-4039 and the lowest values in SPV-11. The highest EC values in water from P-4039 are a consequence of a high concentration of dissolved solids, especially sodium and chloride ions (Table 2a). On the other hand, the water from SPV-11 has the lowest values because of the influence of the Drava River on the alluvial aquifer (dilution effect). The groundwater temperatures ranged from 9.4 to 16 °C. The pH values of groundwater ranged from 6.86 to 7.76, meaning that the waters are mildly acid to alkaline. The DO ranged from 0.2 to 7.1 mg/L. Calculated redox category follows the DO values. Low DO values in the aquifer are accompanied by mixed (oxic-anoxic) conditions and if there is a deficiency in DO, anoxic conditions prevail (Table 3).

The order of dominance ions among cations in waters of observation wells PDS-5, PDS-6, PDS-7, P-1529, P-1556, P-2500, SPV-11 and the private well is $Ca^{2+} > Mg^{2+} > Na^+ > K^+$, while in water of observation well P-4039 is $Na^+ > Ca^{2+} > Mg^{2+} > K^+$ (Table 2a). In a case of anions, the order is $HCO_3^- > NO_3^- > SO_4^{2-} > Cl^-$ in waters of wells PDS-5, PDS-6, PDS-7, P-1529, P-2500 and the private well, but for wells P-1556, SPV-11 and P-4039, it differs. Nitrate concentration in wells PDS-5, PDS-6, PDS-7, P-1529, P-2500 and the private well exceed the maximum contaminant level (MCL) of 50 mg/L $NO_3^-$ [36] for most of the observed time due to agricultural practices and waste water from surrounding settlements. The order of dominance ions among anions for wells P-1556 and SPV-11 is $HCO_3^- > SO_4^{2-} > NO_3^- > Cl^-$, and for well P-4039 is $HCO_3^- > Cl^- > SO_4^{2-} > NO_3^-$. $SO_4^{2-}$ concentrations did not exceed MCL of 250 mg/L in the analyzed samples from all wells, but $Cl^-$ concentrations in water from P-4039 occasionally exceed MCL value of 250 mg/L because of the seasonal de-icing of roads, and sewage waters from semipermeable septic tanks (Table 2a).

Concentrations of nitrite and ammonia in all samples did not exceed MCL values of 0.5 mg/L, ranging from below detection limit to 0.24 mg/L for $NO_2^-$, and from below detection limit to 0.16 mg/L for $NH_4^+$ (Table 2a). Orthophosphate concentrations occasionally exceed MCL value of 0.3 mg/L in waters from wells P-1529, P-1556, P-2500, PDS-5 and PDS-7 (Figure 2).

Table 2b and Figure 3 show that water samples from wells P-1556, P-4039 and PDS-7 contain high concentrations of some heavy metals, occasionally exceeding MCL values. High heavy metal concentrations are attributed to weathering of oxbow sediment, which contains heavy metals, combined with anthropogenic influence. In the rest of the wells, concentrations of heavy metals are very low. Dissolved iron concentrations ranged from 1.1 to 2265 μg/L, manganese concentrations from 0.08 to 237 μg/L, and zinc concentrations from 3.5 to 5665 μg/L (Table 2b and Figure 3). In addition, the waters of well P-4039 show high concentrations of Pb and Ni (Table 2b).

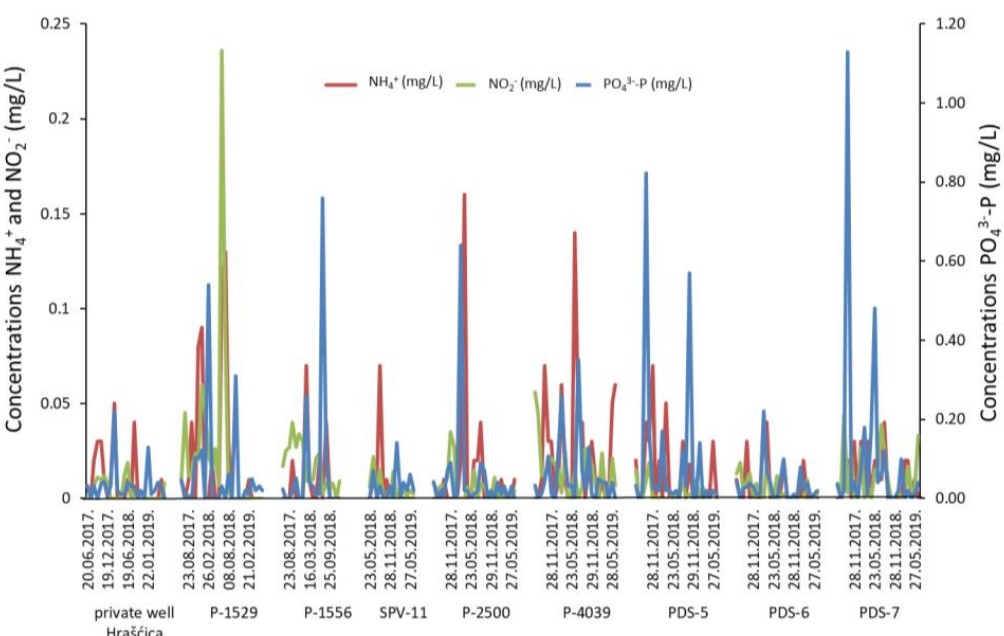

**Figure 2.** Distribution of ammonia, nitrite and orthophosphate in sampled waters by wells.

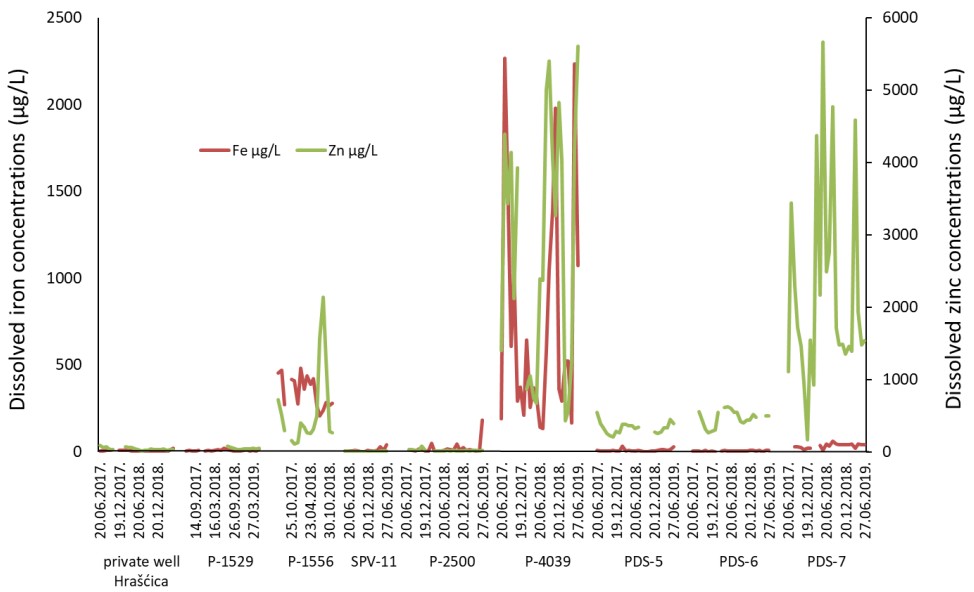

**Figure 3.** Distribution of dissolved iron and zinc concentrations in sampled waters by well.

## 5. Discussion

According to the major ionic composition, sampled waters belong to the CaMg–HCO$_3$ hydrochemical type, except the water from observation well P-4039, which belongs to the NaCa-HCO$_3$ hydrochemical type (Figure 4). Such hydrochemical type of water is a consequence of the dissolution and weathering of carbonate (limestone, dolomite) and silicate minerals (micas, feldspar, etc.) that build aquifer sediments. Since weathering rates of limestone and dolomite are up to 80 and 12 times faster than silicate weathering rates [37], carbonate dissolution mainly dominates major ionic composition and presents the first geochemical process. The influence of silicate weathering, which is the second geochemical process, was analyzed by the bivariate mixing diagram of Na$^+$-normalized Mg$^{2+}$ versus Na$^+$-normalized Ca$^{2+}$ (Figure 5). There is no pronounced silicate weathering in the studied waters, but it was observed that the catchment areas of wells P-4039, P-2500 and occasionally P-1529 and P-1556 indicate the influence of silicate minerals weathering.

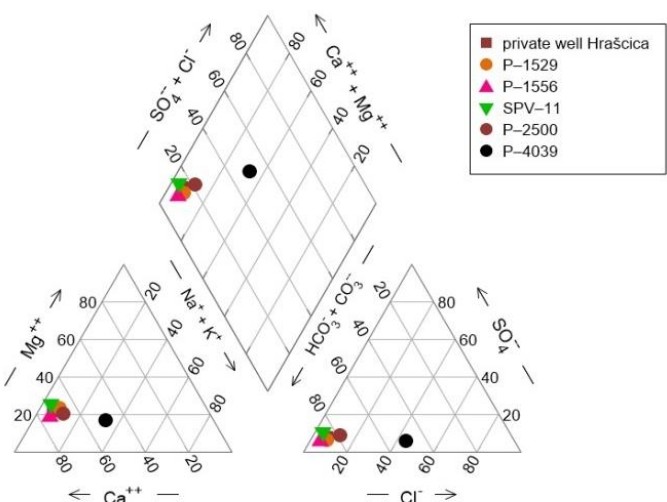

**Figure 4.** Piper diagram of sampled groundwater.

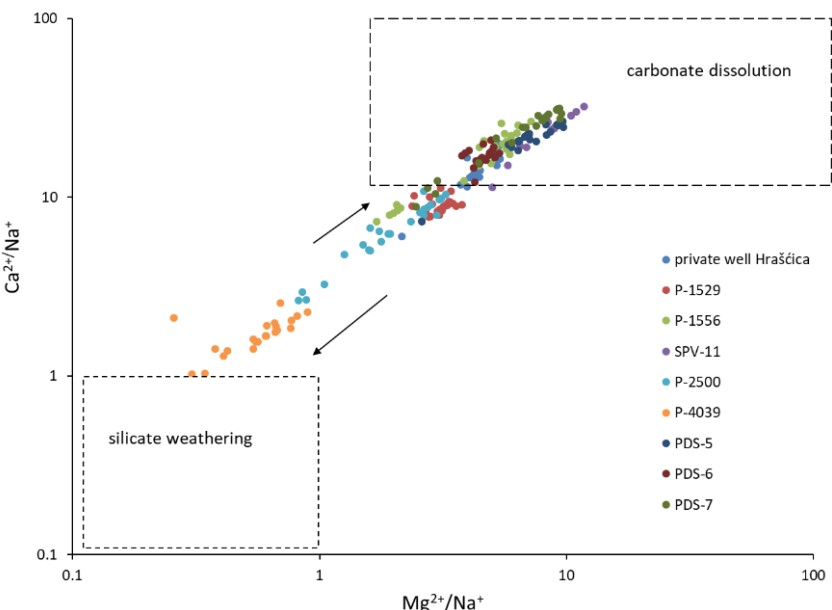

**Figure 5.** Bivariate mixing diagram of $Na^+$-normalized $Mg^{2+}$ versus $Na^+$-normalized $Ca^{2+}$.

In order to determine which carbonate mineral predominantly weathers, molar ratio $Mg^{2+}/Ca^{2+}$ was used. Overall, it is observed that calcite dissolution is dominant over dolomite dissolution (Figure 6). However, in the catchment areas of wells SPV-11, P-4039, P-1529 and PDS-5, the dolomite dissolution is more pronounced than the calcite dissolution because the $Mg^{2+}/Ca^{2+}$ ratios are over 0.33 value (Figure 6).

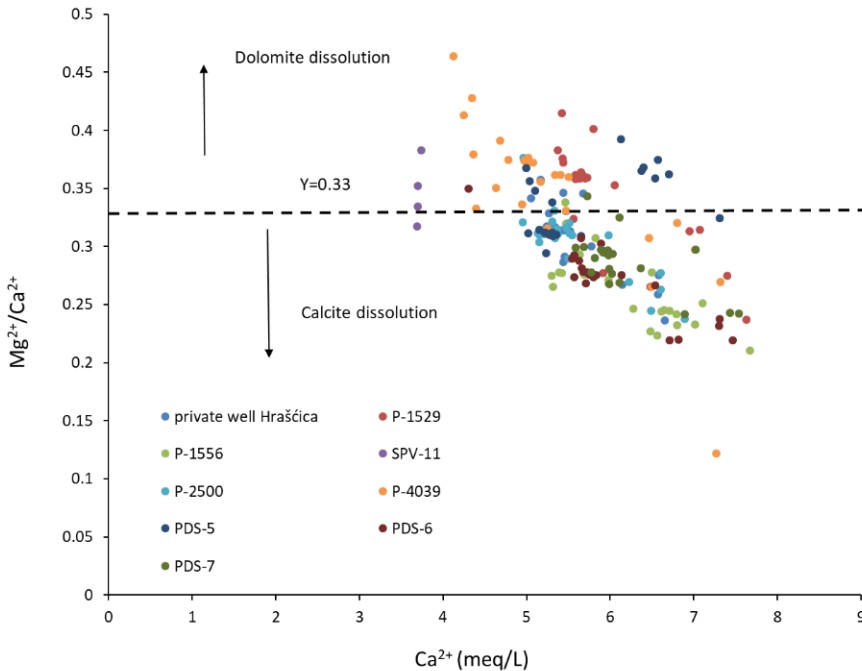

**Figure 6.** Molar ratio $Mg^{2+}/Ca^{2+}$ versus $Ca^{2+}$.

The third geochemical process, cation exchange, was observed, taking into account the bivariate diagram of $(Ca^{2+} + Mg^{2+}) - (HCO_3^- + SO_4^{2-})$ versus $Na^+ - Cl^-$ (Figure 7). Concentrations of bivalent cations ($Ca^{2+}$ and $Mg^{2+}$) that may have been involved in exchange reactions were corrected by subtracting equivalent concentrations of associated anions ($HCO_3^-$ and $SO_4^{2-}$) that would be derived from other processes (e.g., carbonate or silicate weathering, where calcite and anorthite (Ca-feldspar) produce similar molar concentrations of $Ca^{2+}$ and $HCO_3^-$, but no $SiO_2$ [38,39]). Similarly, $Na^+$ that may be derived from the aquifer matrix can be accounted for by assuming that $Na^+$ contributions of meteoric origin would be balanced by equivalent concentrations of $Cl^-$ [40]. For active cation exchange taking place in the aquifer, the slope of this bivariate plot should be $-1$ [41]. Since the cation exchange process is well-pronounced in the catchment of well P-4039 and in the catchments of other wells is masked, two subfigures are given: Figure 7a showing all wells, and in Figure 7b, well P-4039 is left out. It is observed that the cation exchange process is not pronounced in the catchment of the rest of the wells (Figure 7b).

Mineral equilibrium calculations for groundwater are useful in predicting the presence of reactive minerals in the groundwater system and estimating mineral reactivity. By using the saturation index approach, it is possible to predict the reactive mineralogy of the subsurface from groundwater data, without collecting the samples of the solid phase and analyzing the mineralogy [42]. This approach was used, and the saturation indices (SI) of calcite and dolomite and partial pressure of $CO_2$ were calculated. If the groundwater is saturated (SI > 0) with respect to the calcite and/or dolomite minerals, precipitation of calcite and dolomite minerals is possible. On the other hand, if the groundwater is undersaturated (SI < 0) with respect to minerals, dissolution would continue. Most of the time, the sampled groundwater is saturated with respect to calcite and undersaturated with respect to dolomite (Figure 8). Occasionally, especially during summer periods when water levels are decreasing, groundwater is saturated with respect to dolomite. Currently, partial pressure of $CO_2$ is very low and enables precipitation. On the other hand, when water levels increase due to the rainy season, partial pressure of $CO_2$ increases due to the flushing of the surface and unsaturated zone. The SI of both minerals decreases and becomes negative for dolomite and lower or negative for calcite.

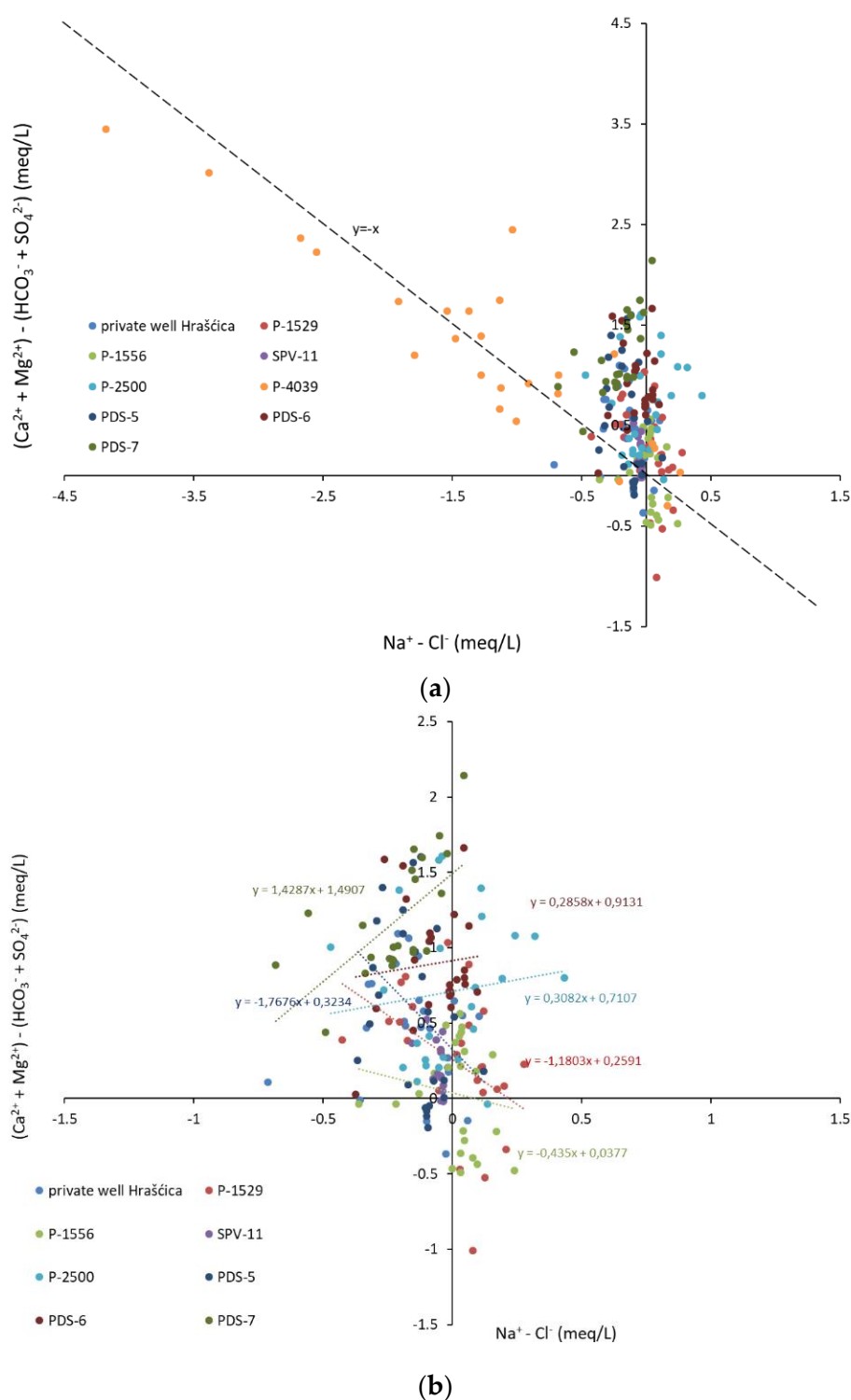

**Figure 7.** Bivariate diagram of $(Ca^{2+} + Mg^{2+})$ - $(HCO_3^- + SO_4^{2-})$ versus $Na^+ - Cl^-$: (**a**) all wells; (**b**) all wells except well P-4039.

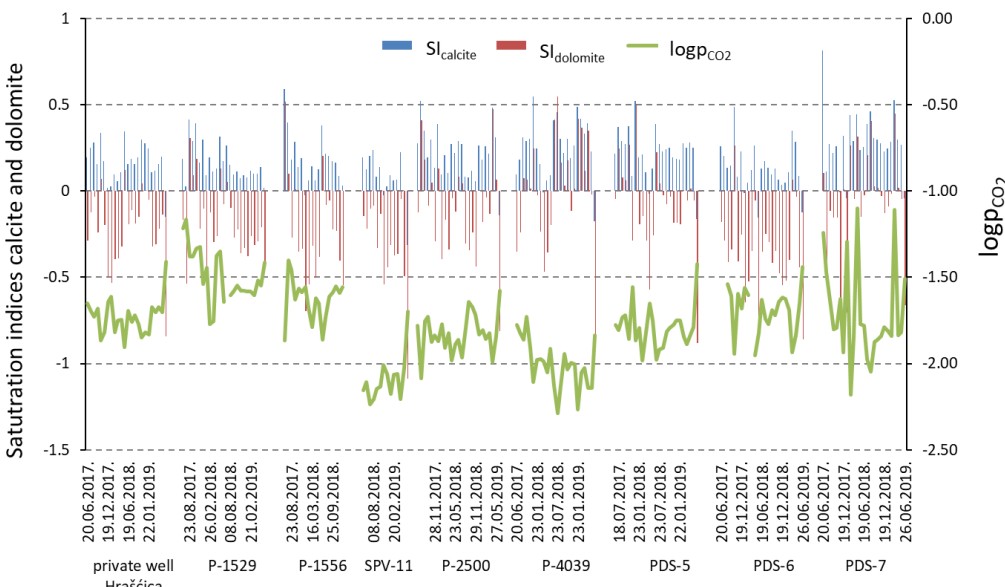

**Figure 8.** Monthly variation of calcite and dolomite saturation indices and partial pressure of $CO_2$ in groundwater.

Higher partial pressure of $CO_2$ is connected with an increase in DOC in groundwater (Figure 9). In catchments of the observation wells P-1529, PDS-7, PDS-5, and PDS-6, it is observed that higher DOC concentrations in water are accompanied by higher partial pressure of $CO_2$, as a consequence of the flushing of the organic matter from the soil and unsaturated zone into the aquifer. However, in the catchments of the observation wells SPV-11, P-1556, and P-2500, changes of DOC concentrations do not significantly affect the partial pressure of $CO_2$, which is mainly controlled by the dissolution of carbonate minerals. In the aquifers with carbonate matrix, it is observed that increasing chemical weathering of carbonate minerals is related to increasing $CO_2$ in groundwater [43].

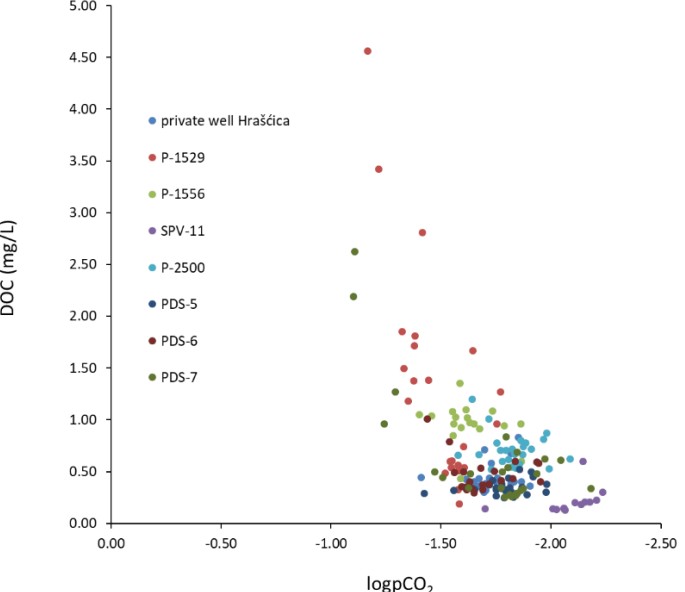

**Figure 9.** Relationship between concentrations of DOC and $logpCO_2$ in sampled waters.

The fourth process that influences the geochemical evolution of groundwater is anthropogenic influence, which is recognized through agricultural and urban activities. In Figure 10, $Na^+/Cl^-$ ratios show that values are mostly scattered around halite line and the

most of samples are shifted to Cl$^-$ side, indicating the influence of the waste water and manure. When cation exchange process is dominant, the values shift to the Na$^+$ side. The source of the halite in the study area is also not natural, but anthropogenic. Halite is used during the winter period for de-icing of the roads.

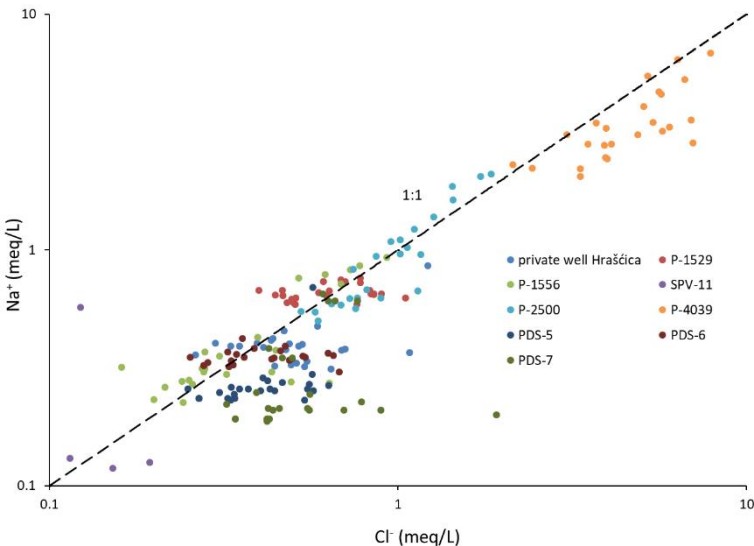

**Figure 10.** Na$^+$ vs. Cl$^-$ in the sampled groundwater.

Another indicator of anthropogenic influence is high nitrate concentration (Figure 11). Usually, the sources of nitrate are fertilizers, organic and mineral [44–46]. Although agricultural production is dominant in the research area, there has been a decrease in agricultural surfaces and the application of fertilizers in the past 10–15 years, followed by an increase in the urban area by 12% [47]. The construction of the sewerage network did not follow the urbanization of the study area, and nitrate pollution may also occur due to the discharge of waste water into the ground. In the catchments of observation wells that are situated in or close to urban area, a positive correlation between nitrate and chloride and a good connection between nitrate and phosphates was observed (Table 2a). It is generally known that source of phosphates is waste water [48]. In addition, bromides were observed in waters of those observation wells, especially during the wet period of the year, and Br$^-$/Cl$^-$ ratio confirms the influence of the waste water (Table 2a). The highest concentrations of nitrate are observed in the middle of the study area, where the intensive agricultural production and urban areas exist. The observation wells that are close to the Drava river have low nitrate concentrations, because the river recharges the alluvial aquifer [9,30] and causes a dilution effect. Moreover, sessional periodic variations of nitrate and chloride were observed (Figure 11). During the winter season and early spring, high chloride concentrations are measured in groundwater samples due to the de-icing of roads. Conversely, high nitrate concentrations are measured during intensive agricultural production and irrigation, which occurs in the late spring–summer season.

In addition to these two anthropogenic indicators, sulfate is also a relevant indicator because it can be released into the groundwater as part of domestic waste water [49]. However, the highest concentrations in groundwater are usually from natural sources such as gypsum, anhydrite, oxidation of sulfide minerals, etc. [49]. In the research area, the origin of sulfate is natural, because concentrations in all catchments are similar.

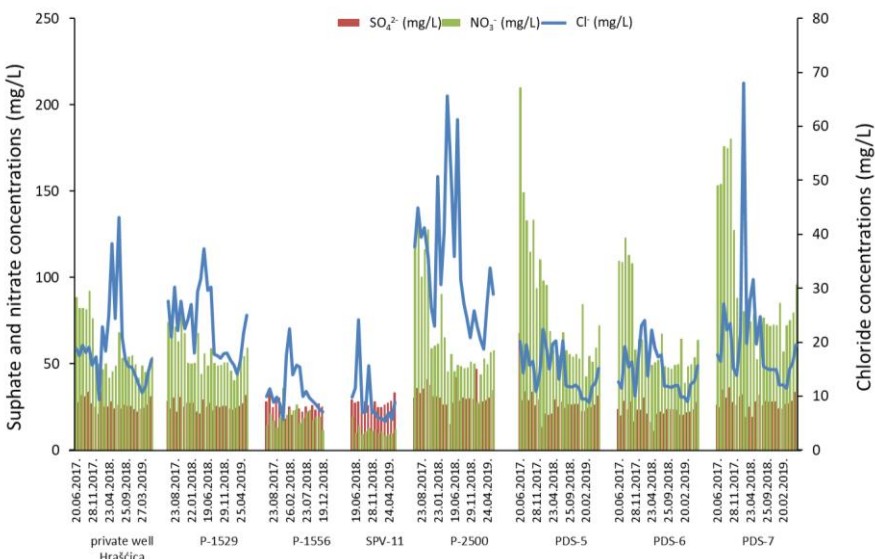

**Figure 11.** Distribution of sulfate, nitrate and chloride in sampled groundwater.

## 6. Conclusions

The alluvial aquifer in the Varaždin region is an important groundwater source for human consumption and the dependent ecosystem. Therefore, it is vital to ensure the sustainable use of this valuable water resource. The conducted research, based on the chemical analyses of groundwater samples from nine observation wells, identified four main processes that influence the groundwater chemistry:

(a) The dissolution and precipitation of carbonate minerals represents the main mechanism controlling the groundwater chemistry. Although the aquifer is composed of carbonate and silicate minerals, carbonate dissolution is dominant against silicate weathering, due to the great difference in their weathering rates. Most of the time, sampled groundwater is saturated with respect to calcite, which enables the precipitation of calcite, and undersaturated with respect to dolomite.

(b) The cation exchange process is well documented in the catchment area of well P-4039, while other observation wells do not show the signs of this process.

(c) The transformation of organic matter is observed in the catchment area of the observation wells P-1529, PDS-7, PDS-5, and PDS-6. High DOC concentrations in water are followed by high partial pressure of $CO_2$, which is a consequence of flushing organic matter from the soil and unsaturated zone into the aquifer.

(d) An anthropogenic influence is recognized through high nitrate concentrations in groundwater. The application of synthetic fertilizers and manure in agricultural production is considered the main source of nitrate contamination. However, changes in land use and recent urbanization caused a more significant impact of waste water on nitrate content in the Varaždin aquifer.

**Author Contributions:** I.K.: investigation, chemical analyses, conceptualization, writing—review and editing; T.M.: funding acquisition, investigation, chemical analyses, conceptualization, data interpretation, writing—original draft preparation; M.Š.M.: chemical analyses; K.M.: chemical analyses and Methods writing. All authors have read and agreed to the published version of the manuscript.

**Funding:** This research was financially supported by the Croatian Scientific Foundation (HRZZ) under grant number HRZZ-IP-2016-06-5365 and by Young Researchers Career Development Project—Training of New PhDs—HRZZ and ESF.

**Data Availability Statement:** The data presented in this study are available in this article. Additional data are available on request from the corresponding author.

**Conflicts of Interest:** The authors declare no conflict of interest.

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
