# Peer review of "Geochemical Characteristics of Alluvial Aquifer in the Varaždin Region"

_water, doi:10.3390/w13111508_

Round 1

Reviewer 1 Report

The aim of the current manuscript is to assess the human and environmental influence on the groundwater chemistry at the Varaždin region. The authors selected an alluvial aquifer and carried out several sampling campaigns from June 2017 to June 2019. The samples were taken from nine observation wells and transported under proper conditions to the laboratory where several analysis were performed such as pH, dissolved oxygen, electrical conductivity, alkalinity, analysis of anions and cations. The results indicate that for specific wells the concentration of some heavy metals occasionally exceed the Maximum Concentration Levels. The dissolution and precipitation of carbonate minerals presents the main mechanism controlling the groundwater chemistry. Furthermore cation exchange process was observed in a single observation well. Finally the authors suggest that the influence of anthropogenic actions was also observed in the region through increased nitrate levels.

The current work can be published in Water because it includes experiments on real world cases and the authors through their analysis assess the human and environmental influence in groundwater quality.

A minor revision is suggested as follows:

  1. Line 57: "(14/10/2013)". The meaning and necessity of the date is not clear. Authors need to clarify it.
  1. Line 124: "NH4+, NO2-  ". The valences +,- should be superscripts of the substance chemical type and not subscripts.
  1. Lines 153-155: "The EC values ranged from 490 to 1091 μS/cm, with highest values measured in water samples from P-4039 and the lowest values in SPV-11." The authors should discuss this variation.
  1. Lines 164-166: "Nitrate concentration in wells PDS-5, PDS-6, PDS-7, P-1529, P-2500 and private well exceed the maximum contaminant level (MCL) of 50 mg/L NO3- [36] for most of the observed time."  Please explain why.
  1. Lines 169-170: "…..Cl- concentrations in 169 water from P-4039 occasionally exceed MCL value of 250 mg/L". Relevant discussion should be added.
  1. Lines 182-183: "Table 2b and Figure 3 show that water samples from wells P-1556, P-4039 and PDS-182 7 contain high concentrations of some heavy metals, occasionally exceeding MCL values." Please explain why.
  1. The spatial variation in the groundwater chemistry is analyzed and discussed. However looking Figure 2, I would suggest if possible, some discussion on temporal variation to be added. Maybe a seasonal periodic variation exists in nitrate (or any other substance) concentrations.

Author Response

Dear Reviewer,

Thank you for your comments and suggestions on the paper. The details of the revisions and responses to your comments follow below.

The aim of the current manuscript is to assess the human and environmental influence on the groundwater chemistry at the Varaždin region. The authors selected an alluvial aquifer and carried out several sampling campaigns from June 2017 to June 2019. The samples were taken from nine observation wells and transported under proper conditions to the laboratory where several analysis were performed such as pH, dissolved oxygen, electrical conductivity, alkalinity, analysis of anions and cations. The results indicate that for specific wells the concentration of some heavy metals occasionally exceed the Maximum Concentration Levels. The dissolution and precipitation of carbonate minerals presents the main mechanism controlling the groundwater chemistry. Furthermore cation exchange process was observed in a single observation well. Finally the authors suggest that the influence of anthropogenic actions was also observed in the region through increased nitrate levels.

The current work can be published in Water because it includes experiments on real world cases and the authors through their analysis assess the human and environmental influence in groundwater quality.

A minor revision is suggested as follows:

  1. Line 57: "(14/10/2013)". The meaning and necessity of the date is not clear. Authors need to clarify it.

The text is modified to clarify the meaning of the date (v. 69).

  1. Line 124: "NH4+, NO2-  ". The valences +,- should be superscripts of the substance chemical type and not subscripts.

Thank you for this insight. It was an error while typing. It is corrected within the text (v. 136).

  1. Lines 153-155: "The EC values ranged from 490 to 1091 μS/cm, with highest values measured in water samples from P-4039 and the lowest values in SPV-11." The authors should discuss this variation.

It is explained according to your suggestion (v. 169-172).

  1. Lines 164-166: "Nitrate concentration in wells PDS-5, PDS-6, PDS-7, P-1529, P-2500 and private well exceed the maximum contaminant level (MCL) of 50 mg/L NO3- [36] for most of the observed time."  Please explain why.

It is explained according to your suggestion (v. 185).

  1. Lines 169-170: "…..Cl- concentrations in 169 water from P-4039 occasionally exceed MCL value of 250 mg/L". Relevant discussion should be added.

It is added according to your suggestion (v. 189-190).

  1. Lines 182-183: "Table 2b and Figure 3 show that water samples from wells P-1556, P-4039 and PDS-182 7 contain high concentrations of some heavy metals, occasionally exceeding MCL values." Please explain why.

It is explained according to your suggestion (v. 205-206).

  1. The spatial variation in the groundwater chemistry is analyzed and discussed. However looking Figure 2, I would suggest if possible, some discussion on temporal variation to be added. Maybe a seasonal periodic variation exists in nitrate (or any other substance) concentrations.

               Thank you for this insight. It was added according to your suggestion (v. 312-316).

Reviewer 2 Report

The manuscript illustrates the hydrochemistry of the alluvial aquifer along with its controlling factors. The manuscript follows the scope of the journal and would be interesting for the readers. However, significant revision is needed to address some important points before its consideration for possible publication. The following comments would help the authors;

  • The aim of the paper is to evaluate environmental and human influences on water chemistry, but no environmental concerns or issues have been discussed in the “results” section. Some recent publications can refer to international journals, e.g. (Environment Earth Sciences, 80: 300. https://doi.org/10.1007/s12665-021-09579-6.);
  • Please highlight the industrial zones in the study area map; also identify which kind of industries have been installed in the region. Does chemistry from sampled water correlate with them?
  • Explain more the protocol of the experiment so that one can repeat this type of work.
  • Explain how trilinear and bivariate diagrams work to illustrate the hydro-chemical facies and their controlling drives respectively. Also specify the protocol to develop these diagrams.
  • Please make sure all the aims and objectives have been achieved in the “results and discussion” part.
  • In the last conclusion, the authors stated the impact of land-use changes on water, but information regarding land-use changes have not been added anywhere in the manuscript.

Abstract:

  • Avoid general statements, be very specific in this part.
  • Please summarize your methodology.
  • Add your key findings.
  • It is more appropriate to add your aims and objectives in the “introduction” part, instead in the “abstract”.
  • Avoid repetition, lines 18, and 19 giving the same idea as in lines 10, and 11.
  • Add take-home message at the end of abstract.

Introduction

Specific:

  • It is too short, please maintain aspect ratio as per total length of the manuscript.
  • Start from the broad area statement and narrow it down to the topic sentence.
  • Add background research regarding natural and man-made influences on hydrochemistry.
  • Add problem statement and importance of the work.
  • Add clear aims and objectives at the end of this part.
  • Page 1-line 43; Do “poultry and dairy farming” are included in agriculture? Please also specify type of industries.

General:

It must contain background, in which problems should be proposed; the present state of the art on the research of the problem, the gap of the present research and the topics to investigate, and objectives of the present research. Length of the Introduction should be 1/8~1/10 (one-eighth to one-tenth) of the paper.

Study Area

  • Page3-line 61; “The study area is the former”; What does it mean?
  • Page3-line 93; the sentence “The covering layer of the aquifer is not continuously developed”, does not make complete sense.
  • Page3-line 99; authors stated “The general groundwater flow direction is NW-SE”, the information does not correlate with Fi. 1.
  • The manuscript focuses on geochemical characteristics, I would suggest to add geological map of the study area.

Materials and Methods:

  • Please add more detail regarding analytical laboratory testing.
  • Add statistical approaches used for this research.
  • Specify tools used to create trilinear piper plots and bivariate diagrams along with procedure.

Results and Discussion:

  • Graphical representation could be improved. Most of the graphs are cluttered and do not represent the data well.
  • Page7-line 174; Please note units for orthophosphate are different in the text and in Fig. 2. Use same unit in the text, as well as in figures for physiochemical parameters.
  • Page9-line 208; “dominant” would be more appropriate word instead “emphasized”
  • Authors mentioned third process (Line~214) and fourth process (Line~264) in the discussion part, but first and second are missing in the part? Please create effective logical flow within text.
  • Discuss all new findings and compare them with previous research to support your results.

Conclusions:

The conclusions should be summarizing the innovative points of the new findings after research. The length of conclusions should be controlled within 1/15~1/20, for the manuscript.

Scientific writing:

The scientific writing of the manuscript requires significant revision. I would like to suggest the manuscript to be professionally proofread and edited. Moreover, the authors may pay attention to some aspect of the conventional research writing, especially the connection between the sentences, the components/structure of the key parts (Abstract, Introduction, body, Conclusion). I suggest the authors reading the following references to learn more about the scientific writing and modify the paper accordingly.

  1. Glasman-Deal, H. (2010). Science Research Writing for non-native speakers of English. Imperial College Press, London, 228p.

Author Response

Dear Reviewer,

Thank you for your comments and suggestions on the paper. The details of the revisions and responses to your comments follow below.

The manuscript illustrates the hydrochemistry of the alluvial aquifer along with its controlling factors. The manuscript follows the scope of the journal and would be interesting for the readers. However, significant revision is needed to address some important points before its consideration for possible publication. The following comments would help the authors;

The aim of the paper is to evaluate environmental and human influences on water chemistry, but no environmental concerns or issues have been discussed in the “results” section. Some recent publications can refer to international journals, e.g. (Environment Earth Sciences, 80: 300. https://doi.org/10.1007/s12665-021-09579-6.);

Environmental concerns are not topic of this study. We have read above mentioned paper, and by „environmental“ we mean natural mechanism, as the authors use this term in the paper about groundwater hydrochemistry in Jhelum Basin. We have further explained it in the abstract (v. 21), to avoid confussion of the reader.

Please highlight the industrial zones in the study area map; also identify which kind of industries have been installed in the region. Does chemistry from sampled water correlate with them?

The industrial zones are not relevant in this study, as they are located at the edge of the study area in the Varaždin City. For this reason, we do not have observation wells downstream to draw any conclusions about correlation between groundwater chemistry and industry. The text about industry is removed from the manuscript.

Explain more the protocol of the experiment so that one can repeat this type of work.

More information added according to your suggestion (v. 130-138).

Explain how trilinear and bivariate diagrams work to illustrate the hydro-chemical facies and their controlling drives respectively. Also specify the protocol to develop these diagrams.

Diagrams are constructed in MS Excel and this sentence is added to the paper (v. 154-155). We believe that the work is explained quite well for each diagram.

Please make sure all the aims and objectives have been achieved in the “results and discussion” part.

Done.

In the last conclusion, the authors stated the impact of land-use changes on water, but information regarding land-use changes have not been added anywhere in the manuscript.

This information is present in the v. 274-277 of the first version of the manuscript (Jogun et al., 2017).

Abstract:

Avoid general statements, be very specific in this part.

Please summarize your methodology.

Add your key findings.

It is more appropriate to add your aims and objectives in the “introduction” part, instead in the “abstract”.

Avoid repetition, lines 18, and 19 giving the same idea as in lines 10, and 11.

Add take-home message at the end of abstract.

The abstract is modified according to your suggestion.

Introduction

Specific:

It is too short, please maintain aspect ratio as per total length of the manuscript.

Start from the broad area statement and narrow it down to the topic sentence.

Add background research regarding natural and man-made influences on hydrochemistry.

Add problem statement and importance of the work.

Add clear aims and objectives at the end of this part.

Page 1-line 43; Do “poultry and dairy farming” are included in agriculture? Please also specify type of industries.

General:

It must contain background, in which problems should be proposed; the present state of the art on the research of the problem, the gap of the present research and the topics to investigate, and objectives of the present research. Length of the Introduction should be 1/8~1/10 (one-eighth to one-tenth) of the paper.

The introduction is partially modified and the text is added according to your suggestion.

Yes, poultry and dairy farming are included in agriculture. The discussion about industry is removed from the manuscript (explained in the second comment).

Study Area

Page3-line 61; “The study area is the former”; What does it mean?

The terms former and latter are words used to distinguish between two things. Former directs us to the first of these two things, and latter directs us to the second (or last) of them. In this instance, between “warm-temperate climate” or “marine west coast climate.” The study area is the former, meaning warm-temperate climate.

Page3-line 93; the sentence “The covering layer of the aquifer is not continuously developed”, does not make complete sense.

The sentence is modified according to your suggestion (v. 106).

Page3-line 99; authors stated “The general groundwater flow direction is NW-SE”, the information does not correlate with Fi. 1.

Figure 1 illustrates the groundwater flow direction very well. As there are some local changes in the groundwater flow net (mainly derived from drainage of the aquifer into derivation channel), the regional flow is in the SE direction.

The manuscript focuses on geochemical characteristics, I would suggest to add geological map of the study area.

Thank you for suggestion. We discussed this matter with co-authors and decided to leave it this way. The main reason is that the entire study area is one geological unit – Quaternary alluvium (this is explained in the paper). Therefore, it is not important to emphasize geology of the study area with geological map.

Materials and Methods:

Please add more detail regarding analytical laboratory testing.

More information added according to your suggestion (v. 130-138).

Add statistical approaches used for this research.

Specify tools used to create trilinear piper plots and bivariate diagrams along with procedure.

Analysis and diagrams are constructed in MS Excel which is a standard for this type of work. Statictical approach is correlation, this is evident from the diagrams. This sentence is added to the paper (v. 154-155).

Results and Discussion:

Graphical representation could be improved. Most of the graphs are cluttered and do not represent the data well.

Diagram in Figure 2 is modified with a few changes. As for the other diagrams, it was exported this way from MS Excel and added to MS Word, resulting in lower quality (probably). The original diagrams are in better quality, so the final version will also be improved.

Page7-line 174; Please note units for orthophosphate are different in the text and in Fig. 2. Use same unit in the text, as well as in figures for physiochemical parameters.

It was corrected according to your suggestion (v. 194). The units are unified in mg/L.

Page9-line 208; “dominant” would be more appropriate word instead “emphasized”

It was modified according to your suggestion (v. 232).

Authors mentioned third process (Line~214) and fourth process (Line~264) in the discussion part, but first and second are missing in the part? Please create effective logical flow within text.

Thank you for this insight. It is added to the text (v. 220-222).

Discuss all new findings and compare them with previous research to support your results.

This is the first paper to be published of this topic in the study area, so there is nothing to compare it with.

Conclusions:

The conclusions should be summarizing the innovative points of the new findings after research. The length of conclusions should be controlled within 1/15~1/20, for the manuscript.

It was modified according to your suggestion, with broader context why it is important to do this work. We believe that we presented the findings of our research clear.

Scientific writing:

The scientific writing of the manuscript requires significant revision. I would like to suggest the manuscript to be professionally proofread and edited. Moreover, the authors may pay attention to some aspect of the conventional research writing, especially the connection between the sentences, the components/structure of the key parts (Abstract, Introduction, body, Conclusion). I suggest the authors reading the following references to learn more about the scientific writing and modify the paper accordingly.

Glasman-Deal, H. (2010). Science Research Writing for non-native speakers of English. Imperial College Press, London, 228p.

Thank you for your reccomendation. We believe that the paper has been improved thanks to your comments and suggestions. In addition, we have decided to check English at the end of the review process through MDPI English editing service.  

Round 2

Reviewer 2 Report

do not reply the comments completely.